# Identifying Predictors for Minimum Dietary Diversity and Minimum Meal Frequency in Children Aged 6–23 Months in Uganda

**DOI:** 10.3390/nu14245208

**Published:** 2022-12-07

**Authors:** Giulia Scarpa, Lea Berrang-Ford, Maria Galazoula, Paul Kakwangire, Didacus B. Namanya, Florence Tushemerirwe, Laura Ahumuza, Janet E. Cade

**Affiliations:** 1School of Environment, University of Leeds, Leeds LS2 9JT, UK; 2School of Food Science and Nutrition, University of Leeds, Leeds LS2 9JT, UK; 3Leeds Institute for Data Analytics, University of Leeds, Leeds LS2 9JT, UK; 4Department of Nutrition, Lira Regional Referral Hospital, Lira P.O. Box 2, Uganda; 5Ministry of Health, Kampala P.O. Box 7272, Uganda; 6Department of Community Health and Behavioural Sciences, Makerere University, Kampala P.O. Box 7062, Uganda

**Keywords:** DHS, Uganda, complementary feeding, minimum dietary diversity, minimum meal frequency

## Abstract

Adequate complementary foods contribute to good health and growth in young children. However, many countries are still off-track in achieving critical complementary feeding indicators, such as minimum meal frequency (MMF), minimum dietary diversity (MDD) and minimum acceptable diet (MAD). In this study, we used the 2016 Ugandan Demographic Health Survey (UDHS) data to assess child feeding practices in young children aged 6–23 months. We assess and describe complementary feeding indicators (MMF, MDD and MAD) for Uganda, considering geographic variation. We construct multivariable logistic regression models—stratified by age—to evaluate four theorized predictors of MMF and MDD: health status, vaccination status, household wealth and female empowerment. Our findings show an improvement of complementary feeding practice indicators in Uganda compared to the past, although the MAD threshold was reached by only 22% of children. Children who did not achieve 1 or more complementary feeding indicators are primarily based in the northern regions of Uganda. Cereals and roots were the foods most consumed daily by young children (80%), while eggs were rarely eaten. Consistent with our hypotheses, we found that health status, vaccination status and wealth were significantly positively associated with MDD and MMF, while female empowerment was not. Improving nutrition in infant and young children is a priority. Urgent nutritional policies and acceptable interventions are needed to guarantee nutritious and age-appropriate complementary foods to each Ugandan child in the first years of life.

## 1. Introduction

Appropriate feeding practices are required for infants and young children to ensure good health, growth and development [1]. While in the first six months of life the World Health Organization (WHO) recommends exclusive breastfeeding, the introduction of complementary foods is recommended from six months onwards for every child [2]. Ensuring adequate nutrition is a key pillar to achieving the right of every child to sufficient nutrition, as laid out by the Convention of the Rights of the Child [3]. However, appropriate and sufficient complementary feeding remains a particular challenge in Sub-Saharan Africa [4].

Undernutrition is still an underlying cause of child mortality and morbidity for children under 5 years globally [5]. In Uganda, rates of undernutrition and underweight have declined since 2000 according to the last available Uganda Demographic and Health Survey (2016). Undernutrition has decreased from 45% in 2000 to 29% in 2016 in Uganda, and underweight from 18% in 2000 to 11% in 2016 [6]. Despite this, high rates of both types of malnutrition persist in the country, together with wasting (4% in 2016) and overweight (4% in 2016) cases among children under five years [6]. To end malnutrition by 2030, the Uganda Nutrition Action Plan 2018–2025 includes in the priority actions the promotion and support of breastfeeding and complementary feeding practices policies and programmes [7].

Recent research has identified potential determinants of adequate complementary feeding, highlighting the role of social and health factors in driving parental choices and ability to support complementary feeding [8,9]. Studies report that the health status of the child is one of the main elements that needs to be considered at the individual level when assessing complementary feeding practices [5]. However, family and social contexts, and especially maternal factors, also influence child nutrition [10]. Maternal education, female empowerment, number of antenatal visits, and household wealth have also been associated with complementary feeding practices [11]. For example, greater female empowerment was found to be associated with reduced wasting in a randomised control trial in Burkina Faso [12]. Findings in Sub-Saharan Africa, however, are mixed, with both increases and decreases in complementary feeding indicators associated with female empowerment [13].

Studies assessing the factors that affect complementary feeding practices in Uganda are limited. A study published in 2017, analysing UDHS data collected in 2006 and 2011, assessed child attributes, but did not explore socio-economic or environmental determinants [14]. Other research has explored complementary feeding practices in specific districts or areas of Uganda [15,16] without assessing predictors of child feeding. Some studies that focused on determinants of infant and young child feeding have been conducted in specific Ugandan regions [17,18,19]. To date, we are aware of no research at the national level assessing the combination of socio-economic, child, and family determinants of child complementary feeding in Uganda.

In this manuscript, we draw on the Ugandan Demographic Health Surveys (UDHS) to assess data collected in 2016—the most recently available DHS data for Uganda—on child feeding practices in young children aged 6–23 months. The aim of this study is to identify priority areas that could be targeted by national authorities, health organizations and the Ugandan Ministry of Health to improve nutrition and complementary feeding practices in young children in Uganda. Our objectives were: 1. To describe complementary feeding indicators in Uganda, considering geographic variation; 2. To assess key determinants previously identified in the literature (empowerment score, wealth index, vaccination and health status) of selected complementary feeding indicators (MMF, MDD) in 2016.

## 2. Materials and Methods

### 2.1. Data Source

The sampling used in the most recent UDHS (2016) comes from the Uganda National Population and Housing Census (NPHC) undertaken in 2014. It followed a two-stage sampling design, including initially 697 clusters (165 urban and 535 rural) from NPHC 2014, and then a sample of 20,880 households from April to October 2016. One hundred and twelve districts were included in the sample, grouped into 15 regions [6].

The data used for this study include socio-economic and demographic information from the Household Questionnaire, and on maternal and child health, nutrition and complementary feeding from the Women’s Questionnaire [6]. The response rate for the household questionnaire was 98% and for the women’s questionnaire was 97%. For this work, we created a subset of the existing UDHS dataset including women with children aged 6–23 months only.

The maps were created through the QGIS program, using the Ugandan subnational administrative boundaries data, and shapefiles for the Lake Victoria [20,21].

### 2.2. Indicators of Complementary Feeding

The World Health Organisation has identified indicators to assess complementary feeding, including introduction of solid, semi-solid or soft food at 6–8 months, MMF, MDD, and MAD [22]. The description of these indicators can be found in Table 1 which refers to 2010 WHO guidelines [22] used in the 2016 UDHS report.

Minimum number of meals recommended for breastfed children is two, and for non-breastfed is four. The 7 food categories used for the MDD indicators are those identified by WHO [23]: 1. grains, roots, tubers and plantains; 2. pulses (beans, peas, lentils), nuts and seeds; 3. dairy products (milk, infant formula, yogurt, cheese); 4. flesh foods (meat, fish, poultry, organ meats); 5. eggs; 6. vitamin-A rich fruits and vegetables; and 7. other fruits and vegetables.

### 2.3. Descriptive Analysis of Complementary Feeding Indicators

We used descriptive statistics to characterize the prevalence of complementary feeding practices in children aged 6–23 months, stratifying by age. We also mapped geographic variation at the district level for these indicators in Uganda.

We additionally summarized key characteristics of the child, mother, father, and household which are considered proximal factors determining complementary feeding practices [5,24]. Specifically, for the child we report information on health and vaccination status, breastfeeding patterns, and other socio-demographic data, including perceived birth weight and birth interval/order. Maternal characteristics comprised data related to birth, number of antenatal visits, education and information level, female empowerment, and domestic violence. We created a score for female empowerment (Appendix A) following previous research criteria [25], and we assigned four different levels based on the resulting distribution (Level 1 = very low; Level 2 = low; Level 3 = medium; Level 4 = high). Paternal characteristics included age, education level and type of occupation; household information referred to the sex of the household, water and toilet condition, and the wealth index composed by UDHS researchers using a principal components analysis method [26].

Other data, which can be included in the societal level of factors potentially influencing complementary feeding practices, related to the geographical regions and the type of areas where the participants involved in the survey lived (urban or rural).

### 2.4. Multivariable Analysis of the Determinants of Complementary Feeding

We used multivariable logistic regression to assess key theorized predictors of complementary feeding. For multivariable analysis, we focus on two outcome variables: MMF and MDD only. Meal frequency and dietary diversity are two of the four main indicators used to assess complementary feeding (CF) practices. Meal frequency is a proxy of individual food security, and it indicates the number of meals received per day; dietary diversity refers to the number of different food groups consumed daily [22]. Evidence suggests that both indicators have an impact on child nutritional and health status [27]; indeed, adequate quality and quantity of foods are linked to better health and nutrition outcomes [28,29]. We could not evaluate the MAD indicator through a multivariable logistic regression model as the number of Ugandan children reaching the MAD was low (63% between 6 and 11 months, and 0% from 12 to 23 months).

### 2.5. Selection of Predictors

To select specific predictor variables, we were guided by a conceptual framework highlighting multi-level determinants of complementary feeding, adapted for Ugandan communities by Scarpa et al. [30] and informed by previous work [5,10,24]. We focused our assessment on three mechanisms theorized as drivers of complementary feeding: child health, household wealth, and female empowerment. Research demonstrates that nutritional status could deteriorate rapidly when a child is sick or unhealthy; in fact, if adequate nutritional intake is not reached, the child’s body is not able to build the immune response, therefore infections and stunting are more likely to occur [31,32]. A particular aspect of child health which may affect complementary feeding is vaccination. Previous research conducted in Uganda with 2006 and 2011 UDHS data showed that administration of DPT3 and measles vaccines were associated with better complementary feeding outcomes [14]. Family or household wealth is widely associated with complementary feeding practices in many studies [33,34,35]. Additionally, according to the literature, wealthier households are likely to have less undernourished children [36,37]. Previous research has demonstrated that female (specifically maternal) empowerment has an effect on complementary feeding practices, but this trend is not seen consistently across research conducted in low-income countries [38]. However, women are usually responsible for the preparation of food and preservation of local food culture in many contexts [39,40]. For this reason, it is not surprising that a higher level of empowerment is associated with better child nutrition outcomes [41,42,43].

To select predictor variables for inclusion in the model(s), we identified a list of variables that were relevant to our three predictors/mechanisms of interest (child health, family wealth, female empowerment), and available within the UDHS dataset. To account for substantial collinearity between available variable options, we undertook the following process: (1) we tested the univariate association for each candidate predictor variable with our two outcome variables (MMF and MDD); (2) we assessed collinearity and/or correlations between predictor variables to identify clusters of similar and highly collinear predictors; (3) we assessed variance inflation factors (VIFs). We reduced the list of potential predictor variables, excluding predictors with a VIF greater than 5 or factors which were highly (positively or negatively) correlated to avoid collinearity.

### 2.6. Control Variables

We additionally controlled for potential confounding variables. We followed the same procedure for control variables as for predictor variables to select final variables for inclusion, and to avoid collinearity. We controlled for the sex of the child, whether or not the child is currently breastfeeding, number of antenatal visits, maternal education and residential location (urban/rural).

Final variables included in the multivariate logistic regression modes are summarized in Table 2.

### 2.7. Multivariable Model

We constructed multivariable logistic regression models for the two selected complementary feeding indicators (MMF and MDD) separately as dependent variables, using female empowerment, wealth index, health and vaccination status as independent predictor variables. Previous studies using DHS data and initial scoping of our data confirmed that there was a substantial difference in MMF and MDD results across age groups (6–11 months, 12–17 months and 18–23 months) [14]; for this reason, we stratified models by age. We included unadjusted (one predictor variable in the model at a time) and fully adjusted (all predictor and control variables included) models. We reported *p*-values and 95% confidence intervals (C.I.) for our findings. Our hypotheses are summarised in Table 3.

Lastly, we ran sensitivity analyses, using three different methods to calculate the missing values in the dataset: 1. we considered the missing values as zero; 2. we imputed the missing data using the multiple imputation change equation (MICE); 3. we used the complete case-analysis.

We used Python 3.8 to analyse the data, and Excel to generate graphs.

## 3. Results

### 3.1. Sample Characteristics

The sample from the 2016 UDHS dataset included 5485 children aged 6–23 months; demographic and socio-economic factors are described in Table 4 and Appendix A. Seventy-four percent of children were breastfed when the survey was conducted. Fewer than half (45%) were the second to fourth child in the family, with the majority (59%) born two or more years after the previous child. While 60% of the children were reported to have received vitamin A and 40% were fully vaccinated, fewer than 10% of children received iron supplementation. More than half of the children sampled (60%) were reported as having been sick in the previous two weeks.

The majority of mothers delivered at health facilities with skilled health professionals (76%) and attended four or more antenatal visits appointments (63%) during pregnancy. More than 60% of mothers had a primary school level of education, 55% used a radio as source of information, and 74% were employed in manual activities. Most mothers scored low to medium for level of empowerment (42%). The majority of fathers attended primary school only (55%) and worked in an agriculture-related field or did other manual jobs (74%).

In nearly all cases (96%) the water source was not in the same dwelling or yard of the household and was reachable within 1 h of distance (56%). Two-third of the families did not have an improved toilet facility (67%) and 81% of lived in a rural area.

### 3.2. Description of Complementary Feeding Foods and Distribution of CF Indicators by Child Age Groups and by Region

A greater proportion of children between 6 and 11 months were found to reach MMF (76%) compared to older children. Conversely, the MAD indicator was met by the youngest age group only (63%). However, the proportion of children meeting the MDD indicator was low among all age groups, ranging from 26% (children 6–11months) to 34% (18–23 months) (Figure 1).

Overall, only one-third of children in the sample consumed four or more different food groups (Figure 1). The most consumed food group included cereals and roots, which accounted for 80% of reported foods by the participants. This was followed by fruits and vegetables (55%), and legumes (45%), showing that the diet was largely vegetarian. Although flesh foods, including meat and other animal products, counted for 30% of eaten foods among the children, meals containing eggs were not frequent (10%). The consumption of sweets was also generally limited (10%), but fats and oil were added to main meals (33%). No substantial differences were found in the consumption of food categories among the three age groups (Appendix A).

Complementary feeding prevalence varied geographically within Uganda, with areas of high and low prevalence inconsistent across indicators (Figure 2A,B). A relatively low percentage of children living in the northern districts of Uganda met three or more complementary feeding indicators, although most of them started complementary feeding at 6 months of life. Conversely, in western and central Ugandan districts, we observed the highest percentage of children achieving three or more complementary feeding indicators, but this trend was not found for Kampala district. Sixty to eighty percent of children in many Ugandan districts reached the introduction of complementary feeding indicator, while the MDD and MAD indicators were only achieved by 20–40% of children in most of the regions.

### 3.3. Determinants of MMF and MDD

We assessed evidence of statistically significant association between our 4 hypothesized predictor variables (female empowerment, wealth index, health and vaccination status) and 2 outcome variables (MMF and MDD), testing each of the hypotheses presented in the methods text (Table 5).

Consistent with our hypotheses, a child who was reported as having been sick in the previous 2 weeks was significantly associated (in both unadjusted and fully adjusted models) with a greater odds of meeting the MMF and MMD thresholds except for the youngest age group (6–11 months). At the 95% confidence level in the fully adjusted model, a child reported as sick in the past 2 weeks, in fact, had a 1.46 (CI 1.14–1.90, *p* ≤ 0.001) (12–17 months) and 1.28 (CI 1.00–1.61, *p* = 0.04) (18–23 months) times greater odds of meeting the MDD threshold, and a 1.48 (CI 1.17–1.87, *p* ≤ 0.001) (12–17 months) and 1.31 (CI 1.05–1.64, *p* = 0.02) (18–23 months) greater odds of meeting the MMF threshold than a child who did not report being sick in fully adjusted models. 

In line with our hypotheses, the vaccination predictor (being partially and fully vaccinated) was significantly associated with both MMF and MDD indicator across all age groups. However, in the stratified models it was a significant predictor of MMF in the fully adjusted models for the 12–17 months age group, and of MDD only in the 6–11 months age group; vaccination was a significant predictor of MMF in unadjusted models only. At the 95% confidence level in the fully adjusted model, a child who was fully vaccinated had a 1.27 (CI 0.76–2.15, *p* = 0.36) (6–11 months) and 4.33 (CI 2.42–7.74, *p* ≤ 0.001) (12–17 months) times greater odds of meeting the MMF threshold, and a 2.86 (CI 1.43–5.71, *p* ≤ 0.001) (6–11 months) and 4.65 (CI 2.15–10.00, *p* ≤ 0.001) (12–17 months) greater odds of meeting the MDD threshold than children who were not vaccinated. In the 18–23 months age group, more than three quarter of children were partially or fully vaccinated, and it was not possible to perform multivariable logistic regression.

Despite mixed results, the wealth index was found to be significantly associated across the majority of models and across all age groups for both MMF and MDD; despite reductions in significance between the unadjusted and full adjusted models, significance largely persisted when accounting for other predictors and control variables. Indeed, at the 95% confidence level in the fully adjusted model, household in the highest wealth percentile had a 1.90 (CI 1.23–2.92, *p* ≤ 0.001) (6–11 months), 2.73 (CI 1.74–4.30, *p* ≤ 0.001) (12–17 months) and 1.80 (CI 1.20–2.77, *p* = 0.01) (18–23 months) times greater odds of meeting the child MDD threshold than a household in the lowest wealth percentile. This was largely consistent across age groups except for mixed results for households at lower levels of wealth for children 12–17 months. The results also indicate greater confidence in the association of higher levels of wealth with the odds of reaching the MDD threshold, and mixed confidence in the role of moderate increases in wealth. Wealth was significantly associated with MMF in unadjusted models but reached significance for only a third of strata in fully adjusted models. Among children 6–11 months and 12–17 months, only the highest quintile of wealth was significantly associated with greater odds of a child reaching the MMF threshold, though we do observe increasing odds at higher levels of wealth with borderline *p*-values (particularly for 6–11 months). Indeed, at the 95% confidence level in the fully adjusted model, household in the highest wealth percentile had a 1.90 (CI 1.23–2.92, *p* ≤ 0.001) (6–11 months), 2.73 (CI 1.74–4.30, *p* ≤ 0.001) (12–17 months) and 1.80 (CI 1.20–2.77, *p* = 0.01) (18–23 months) times greater odds of meeting the child MDD threshold than a household in the lowest wealth percentile.

Female empowerment was not statistically associated with either MMF or MDD in the models for all age groups, but it was in some of the models for the 18–23 months children, particularly at lower levels of female empowerment. The association between female empowerment and MDD reached statistical significance for the middle categories only (‘low’ female empowerment and ‘medium’ female empowerment). Indeed, at the 95% confidence level in the fully adjusted model, a mother who had a level of empowerment equal to ‘low’ and ‘medium’ had a 1.48 (CI 1.15–1.92, *p* ≤ 0.001) and 1.39 (CI 1.04–1.87, *p* = 0.03) times greater odds of meeting the child MMF threshold, and a 1.38 (CI 1.10–1.80, *p* = 0.02) and 1.62 (CI 1.20–2.20, *p* ≤ 0.001) greater odds of meeting the child MDD threshold than a mother who had a very low level of empowerment. The fact that this predictor is significant only in the adjusted models implies that female empowerment interacted and/or was confounded by other variables which had an effect on MMF and MDD.

The sensitivity analysis showed no significant differences in the results when we considered the missing values as zero, and when we imputed the data. However, when we used the complete case analysis with the sample reduced of over 40% there were differences in the OR of the vaccination predictor, which was lower than in the other models. This was due to a different distribution (number of participants) of the variable (Appendix A).

## 4. Discussion

Our study investigated the complementary feeding practices among children aged 6–23 months in Uganda using 2016 UDHS data. Specifically, we assessed the main complementary feeding indicators—including MMF, MDD, MAD—and the introduction of complementary feeding together with the food groups consumed by the participants. Additionally, we analysed the variation in complementary feeding practices across geographical areas, and we tested four predictors (wealth, vaccination, sickness and female empowerment) of complementary feeding to evaluate any association with MMF and MDD indicators.

In general, children who are reported as having been sick in the past 2 weeks, who have received complete age-appropriate vaccinations and who lived in households with higher wealth indices were more likely to have met the standard for MMF and MDD. Female empowerment was not a significant predictor for the selected complementary feeding indicators.

The proportion of children meeting complementary feeding indicators improved compared to the previous UDHS in 2011 (4), especially for the MMF and MDD indicators, but still remained low. A greater proportion of children also started complementary feeding at 6 months of age. Yet, infant and young child feeding practices remain suboptimal across Uganda. This implies that many children are still vulnerable to stunting and micronutrient deficiencies, with a risk of increased morbidity and mortality [1,48]. In fact, the prevalence of stunting remains about 40% in many northern areas of the country [6]. According to our findings, in these same districts the percentage of children meeting the MDD, MMF and MAD was low, although complementary foods were mostly introduced at the advised age of 6 months (Appendix A).

We also explored the most consumed foods during the complementary feeding period, finding similarities with results from elsewhere in Sub-Saharan Africa and Southern Asia [19,49]. The most frequently eaten food groups were cereals and roots, while proteins and foods rich in iron were less frequently consumed. This pattern of consumption was found to be common in many low income settings, and likely to be the cause of protein and iron deficiency in disadvantaged communities [50]. In fact, the quality of the proteins consumed is often low, limiting the availability of the protein for use in the body [50]. More investment in the development of local agriculture interventions is needed to guarantee nutritional well-being [51]. Eggs were rarely eaten by children in the 24 h prior to the survey. Previous studies demonstrated that egg consumption can bolster growth in highly vulnerable periods, and in a 2015 randomized controlled trial consuming eggs reduced stunting by 47% in Ecuador [52,53,54]. Therefore, increasing the consumption of eggs among Ugandan children may be beneficial for their nutritional status. Additionally, eggs and animal protein-based foods may reduce the prevalence of anaemia in Uganda, which is especially high among children 6–23 months old [6].

We found wealth to be a statistically significant predictor of MMF and MDD indicators across all age groups. This may evidence the need of reducing household poverty to address complementary feeding practices as demonstrated by other studies [55,56]. Research has shown that children living in deprived households were associated with an increased risk of inadequate dietary diversity [57]. An unsuitable diet can lead to a greater risk of undernutrition; household wealth inequality has been found to be strongly associated with childhood stunting in other studies [58,59].

Children’s health status was found to be significant for both MMF and MDD indicators in our models. Being sick was associated with the two complementary feeding indicators in the older age groups, but not among the 6–11 months old group in which the prevalence of breastfeeding was higher. This may be due, in the first instance, to the cross-sectional nature of the UDHS dataset. Additionally, parents are more likely to feed a sick child better as reported in other studies conducted in Ethiopia and India [60,61]. In fact, findings showed that mothers increased the frequency and quality of foods when the child was ill [44,45]. Although appetite may be reduced, diverse food is required for a sick child to recover quicker and maintain an adequate nutrient intake [62].

Having completed the relevant course of vaccinations for each age group was also positively associated with the complementary feeding predictors. Mothers who bring children for vaccinations may access information on various health and nutrition subjects, including complementary foods [11,13]. Researchers elsewhere have reported success in integrating nutritional education into vaccinations campaigns [63]; however, literature on this remains limited, and feasibility is context dependent.

Female empowerment variables were not statistically significant predictors of our outcome indicators, apart from the adjusted model for the age group 18–23 months. This is consistent with the results of other studies using DHS data [11,13], and may be due to heterogeneity in describing and defining women’s empowerment [64,65]. In this study, we used a female empowerment score, which has been previously applied in other research studies [13,25]. However, female empowerment is a context-specific process [66]; although there is evidence that it is associated with child nutritional status [67], ways to measure it and associated findings may not be generalizable [38]. In our data, for example, wealthier and urban households reported lower levels of female empowerment than poorer and rural households, suggesting potential interaction and confounding of female empowerment with wealth and geography, and reflecting complex mechanisms and measurement of female empowerment more generally.

### Study Limitations

Some limitations are present in our study. First, we recognize the cross-sectional nature of DHS data, which limited our ability to assess variation over time. Second, there are missing data in some Ugandan districts, and for some ethnicities such as the Batwa Indigenous population, despite well-recognized differences in food systems and food culture by ethnicity [68,69]. More research targeting vulnerable groups is needed to tailor future nutrition interventions to the different Ugandan areas. Third, mothers were interviewed only once about foods and liquids consumed by the children the previous day, so that variation by day of week was not available. Fourth, standardised questions are used in the UDHS without additional consideration or validation within diverse socio-cultural contexts, and not directly observing the feeding practices with the possibility for the participants of over or under-reporting. Fifth, we could not validate the empowerment score on the field due to budget restrictions.

## 5. Conclusions

Our work highlights that a Ugandan child’s health, either being sick, or partially/fully vaccinated and living in households with higher wealth index had higher odds of meeting MMF and MDD indicators compared to children who were well; not vaccinated or living in poorer households. This study calls for urgent nutritional policies on child nutrition, and complementary feeding practices targeting disadvantaged children. Integration of poverty eradication interventions along with health and nutrition education is required to make the nutritional programmes successful. These joint strategies, in fact, would support provision of nutritious and age-appropriate complementary foods to each Ugandan child in the first 1000 days of life, and improve child health and nutrition outcomes in the longer term.

## Figures and Tables

**Figure 1 nutrients-14-05208-f001:**
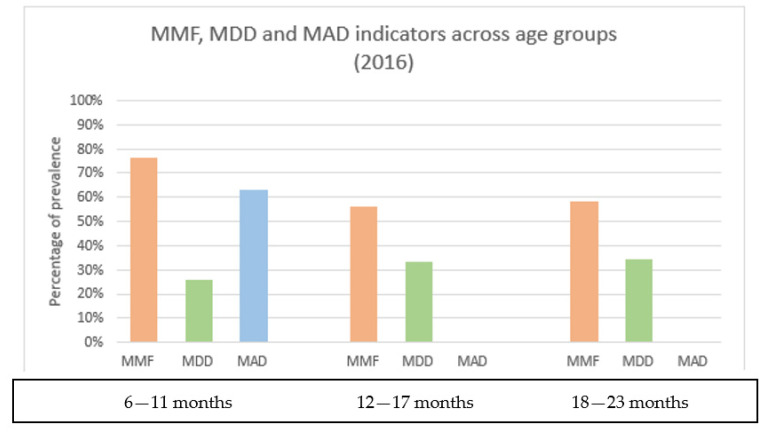
Complementary feeding practices indicators (2016) by child age groups.

**Figure 2 nutrients-14-05208-f002:**
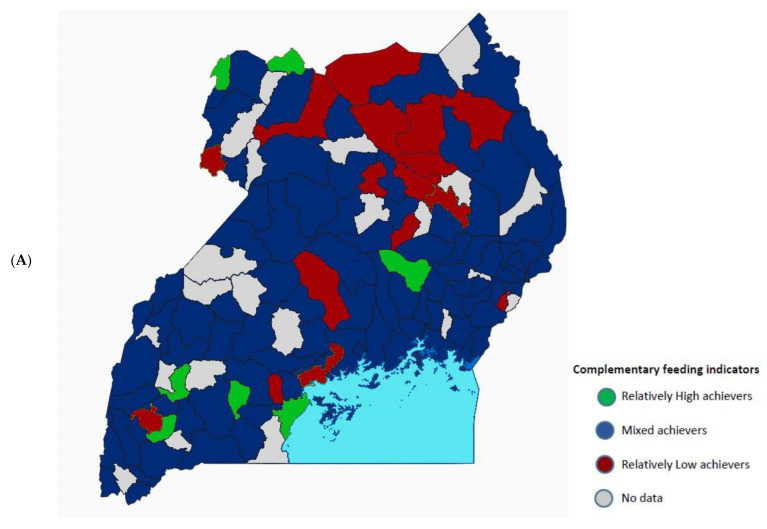
(**A**) Composite map describing the complementary feeding indicators (MMF, MDD & MAD) in Uganda. Group 1, in green, represents those falling into the top quartile of districts (higher%) for 3 or more indicators (relatively high achievers); group 2, in red, represents those falling into the bottom quartile of districts (lowest%) for 3 or more indicators (relatively low achievers); Group 3, in blue, represents all other districts (mixed achievement). (**B**) Geographical distribution of children meeting each complementary feeding indicator in Uganda indicated with different colours (yellow = 81–100%; light green = 61–80%; dark green = 41–60%; blue = 21–40%; purple = 0–20%). Maps created through the QGIS program.

**Table 1 nutrients-14-05208-t001:** Description of complementary feeding indicators used in this study according to WHO guidelines (2010).

Indicators	Indicator Definition	6–8 Months	9–23 Months
Introduction of solid, semi-solid or soft food 6–8 months	Percentage of infants 6–8 months of age who consumed solid, semi-solid or soft foods during the previous day.	Same for all age groups	Same for all age groups
Minimum dietary diversity (MDD)	Percentage of children 6–23 months of age who consumed foods and beverages from at least four out of seven defined food groups during the previous day	Same for all age groups	Same for all age groups
Minimum meal frequency (MMF)	Percentage of children 6–23 months of age who consumed solid, semi-solid or soft foods (also including milk feeds for non-breastfed children) at least the minimum number of times during the previous day.	Breastfed children: Number of solid, semi-solid, or soft foods ≥2 Non-breastfed children: Total of solid, semi-solid, or soft foods AND milk feeds ≥4	Number of solid, semi-solid, or soft foods ≥3
Minimum acceptable diet (MAD)	Percentage of children 6–23 months of age who consumed a minimum acceptable diet (see columns to right for definitions according to age group and breastfeeding status) during the previous day.	Breastfed children: Number of food categories ≥4 AND Number of solid, semi-solid, or soft foods ≥2 Non-breastfed children: Number of food categories ≥4, AND Number of milk feeds ≥2, AND Total number of solid, semi-solid, or soft foods AND milk feeds ≥4	Number of food categories ≥4 AND Number of solid, semi-solid, or soft foods ≥3

**Table 2 nutrients-14-05208-t002:** Dependent and independent variables, control variables and stratification variables used in this study.

Variable	Definition	Description
*Outcome variables*		
Minimum dietary diversity	Did the child consume food and beverages from at least four out of seven defined food groups during the previous day? Binary variable: Yes/No	This variable was also stratified by age groups (6–11 months, 12–17 months, 18–23 months) based on international indicators.
Minimum meal frequency	Did the child eat the minimum number of times which is appropriate for his/her age during the previous day (2, 3 or 4 times depending on breastfeeding status)? Binary variable: Yes/No	This variable was also stratified by age groups (6–11 months, 12–17 months, 18–23 months) based on international indicators.
*Predictor variables*		
Female empowerment	What is the female empowerment score for the mother? Categorical variable: Very low, low, medium, high	This score (0–13 points) was created and previously used as a discrete variable by Jennings et al. [25]; however for this study we grouped the score in 4 categories (very low if the score was lower than 4, low if the score was equal to 5–6, medium if the score was equal to 7–9, high if the score was higher than 10).
Family wealth	What is the wealth percentile of the child’s family? Categorical variable: lowest, second, middle, fourth, highest	The family wealth is a composite variable found in the UDHS 2016, which is calculated based on: house’s ownership; materials used to build the house; typology of sanitation facilities and water access. It was generated using principal components analysis. The variable divides household wealth into 5 wealth quintiles.
Child health—vaccination status	Has the child completed vaccinations for his/her age? Categorical variable: fully vaccinated, partially vaccinated, not vaccinated	Fully vaccinated children included those who received all vaccinations according to their age group (6–11 months, 12–17 months & 18–23 months). Partially vaccinated children included those who received at least 1 vaccination, but not all vaccinations according to their age group. Not vaccinated children included those who did not receive any vaccinations.
Child health—sick in the past 2 weeks	Was the child sick with fever, cough or diarrhoea in the past 2 weeks? Binary variable: Yes/No	This variable is composite, and derives from 3 different variables available in the UDHS 2016: Did the child have fever in the past 2 weeks? Did the child have a cough in the past 2 weeks? Did the child have diarrhoea in the past 2 weeks? A child was considered sick when one or more of these three variables was positive.
*Control variables*		
Sex of child	Which is the sex of the child? Binary variable: Female/Male	Female or male child
Current breastfeeding status	Is the child currently breastfeeding? Binary variable: Yes/No	This variable investigates if the child is still being breastfed and not only if he/she was breastfed.
N. Antenatal visits	How many antenatal visits did the mother attend during pregnancy? Categorical variable: No visits, 1–3 visits, >4 visits	Number of antenatal visits attended during pregnancy by the mother.
Maternal education	What is the mother’s education level? Categorical variable: No education, primary education, secondary education or higher	Highest level of education that the mother acquired in her life, divided into 3 categories: no education, primary education and secondary education/higher.
Geographic location	Do the family live in a city or rural area? Binary variable: urban/rural location	Place of residence: urban or rural. The answer is not formulated by the respondent, but it is defined based on the place where the cluster or sample is based.

**Table 3 nutrients-14-05208-t003:** Hypotheses formulated for this study.

Hypothesis Number	Hypothesis for the MDD Indicator	Hypothesis for the MMF Indicator	Evidence Justifying the Hypothesis
H1	Children who have reported as being sick in the past 2 weeks are more likely to have met the standard for minimum dietary diversity.	Children who have reported as being sick in the past 2 weeks are more likely to have met the standard for minimum meal frequency.	Children, when sick, are more likely to be fed with more and more nutritious food by their mothers [44,45]
H2	Children who have received complete age-appropriate vaccinations are more likely to have met the standard for minimum dietary diversity.	Children from who have received complete age-appropriate vaccinations are more likely to have met the standard for minimum meal frequency.	Mothers attending vaccination clinics get more information on child feeding [11]
H3	Children are more likely to have met the standard for minimum dietary diversity at increasing levels of household wealth.	Children are more likely to have met the standard for minimum meal frequency diversity at increasing levels of household wealth.	Children living in wealthier households are more likely to eat a diverse and balanced diet [46]
H4	Children are more likely to have met the standard for minimum dietary diversity with increasing levels of female empowerment.	Children are more likely to have met the standard for minimum meal frequency with increasing levels of female empowerment.	Mothers with higher level of female empowerment are more likely to have children with better nutritional outcomes [43,47]

**Table 4 nutrients-14-05208-t004:** Sample characteristics of children aged 6–23 months (UDHS 2016).

	Sample *N*	Count (Percentage)
**Child characteristics**		
Child sex	5485	
Female		2711(49%)
Male		2774 (51%)
Breastfeeding status	5485	
Still breastfed		4077 (74%)
Not breastfed		1408 (26%)
Age (in months)	5485	
6–11		1989 (36%)
12–17		1820 (33%)
18–23		1676 (31%)
Completed age-appropriate vaccination	5485	2196 (40%)
Child reported as sick in the past 2 weeks	5148	3112 (61%)
Child health: had the following symptom in the past 2 weeks	5148	
Diarrhoea		1766 (34%)
Fever		2118 (41%)
Cough		2442 (47%)
**Maternal characteristics**		
Antenatal clinic visits	4800	
None		87 (2%)
1–3		1699 (35%)
>=4		3014 (63%)
Highest educational level	5485	
No education		690 (13%)
Primary		3332 (61%)
Secondary or higher		1463 (27%)
Female empowerment—woman involved in decision making on:	5485	
How woman’s income is used		2404 (44%)
How man’s income is used		2313 (42%)
Large household purchases		2896 (53%)
Visiting family and friends		3229 (59%)
Regarding own health care		3334 (61%)
Female empowerment—other		
Woman salary similar/higher than man salary		562 (10%)
Woman owns a land		2062 (38%)
Woman owns a house		2619 (48%)
Attitude towards domestic violence—Beating justified if a woman does the following:	5485	
Goes out without telling him [male/husband]		2151 (40%)
Neglects the children		2247 (61%)
Argues with him [male/husband]		1650 (30%)
Refuses to have sex with him [male/husband]		1092 (20%)
Burns the food		827 (15%)
Women’s empowerment score	5485	
Very low (score 1–3)		1374 (29%)
Low (score 4–6)		2000 (43%)
Medium (score 7–9)		1115 (24%)
High (score 10–13)		218 (5%)
**Household characteristics**		
Household wealth	5485	
Poorest		1536 (28%)
Poorer		1102 (20%)
Middle		1027 (19%)
Richer		908 (17%)
Richest		912 (17%)
Geographical region	5485	
Capital		257 (5%)
Central Uganda		942 (17%)
West Uganda		1379 (25%)
East Uganda		1558 (28%)
North Uganda		1349 (25%)
Type of residence	4707	
Rural		3809 (81%)
Urban		898 (19%)

**Table 5 nutrients-14-05208-t005:** Unadjusted and fully adjusted models for the MMF and MDD indicators for all age groups and stratified by the three age groups 6–11 months, 12–17 months and 18–23 months (a,b). The results in the table were obtained considering missing values as zero. (a) MMF: Unadjusted and fully adjusted models for the MMF indicator for all age groups and stratified by age; (b) MDD: Unadjusted and fully adjusted models for the MDD indicator for all age groups and stratified by age.

**(a) MMF**																
		**All Ages**				**6–11 Months**				**12–17 Months**					**18–23 Months**	
**MINIMUM MEAL FREQUENCY**	**UNADJUSTED**		**ADJUSTED**		**UNADJUSTED**		**ADJUSTED**		**UNADJUSTED**		**ADJUSTED**		**UNADJUSTED**		**ADJUSTED**	
	**OR (95% CI)**	***p* Value**	**OR (95% CI)**	***p* Value**	**OR (95% CI)**	***p* Value**	**OR (95% CI)**	***p* Value**	**OR (95% CI)**	***p* Value**	**OR (95% CI)**	***p* Value**	**OR (95% CI)**	***p* Value**	**OR (95% CI)**	***p* Value**
** *Vaccination status* **																
Not vaccinated	1		1		1		1		1		1		1		1	
Partially vaccinated	2.91 (2.30–3.71)	<0.001	2.21 (1.70–2.88)	<0.001	2.20 (1.41–3.42)	<0.001	1.44 (0.84–2.47)	0.18	5.82 (3.36–10.08)	<0.001	3.40 (1.89–6.10)	<0.001	n/a	n/a	3.39 (2.10–5.59)	n/a
Fully vaccinated	4.04 (3.15–5.19)	<0.001	2.70 (2.04–3.57)	<0.001	2.00 (1.33–3.01)	<0.001	1.27 (0.76–2.15)	0.36	7.51 (4.37–12.90)	<0.001	4.33 (2.42–7.74)	<0.001	n/a	n/a	n/a	n/a
** *Female empowerment* **																
Very low female empowerment	1		1		1		1		1		1		1		1	
Low female empowerment	1.08 (0.94–1.25)	0.13	1.11 (0.96–1.29)	0.16	0.85 (0.64–1.12)	0.24	0.86 (0.65–1.13)	0.28	1.05 (0.82–1.34)	0.69	1.02 (0.79–1.31)	0.91	1.33 (1.05–1.70)	0.02	1.48 (1.15–1.92)	<0.001
Medium female empowerment	1.03 (0.87–1.21)	0.75	1.08 (0.91–1.29)	0.36	0.87 (0.63–1.20)	0.41	0.91 (0.66–1.27)	0.59	1.02 (0.77–1.34)	0.92	1.01 (0.75–1.36)	0.96	1.20 (0.91–1.58)	0.19	1.39 (1.04–1.87)	0.03
High female empowerment	0.84 (0.63–1.13)	0.25	0.90 (0.66–1.21)	0.48	1.05 (0.58–1.89)	0.88	1.12 (0.61–2.06)	0.71	0.88 (0.54–1.43)	0.59	1.02 (0.60–1.72)	0.95	0.65 (0.39–1.09)	0.10	0.64 (0.37–1.10)	0.10
** *Wealth index* **																
First wealth percentile	1		1		1		1		1		1		1		1	
Second wealth percentile	1.34 (1.13–1.59)	<0.001	1.28 (1.07–1.53)	<0.001	1.40 (1.01–1.94)	0.04	1.39 (1.00–1.95)	0.05	1.32 (0.98–1.76)	0.06	1.25 (0.92–1.70)	0.15	1.43 (1.07–1.92)	0.02	1.22 (0.89–1.67)	0.21
Middle wealth percentile	1.30 (1.08–1.55)	0.04	1.24 (1.03–1.50)	0.03	1.49 (1.07–2.08)	0.02	1.44 (1.02–2.03)	0.04	1.10 (0.81–1.49)	0.53	1.06 (0.76–1.46)	0.74	1.38 (1.01–1.89)	0.04	1.20 (0.85–1.67)	0.30
Fourth wealth percentile	1.45 (1.20–1.75)	<0.001	1.43 (1.17–1.76)	<0.001	1.53 (1.07–2.20)	0.02	1.44 (0.98–2.13)	0.06	1.38 (1.00–1.89)	0.05	1.40 (0.99–1.99)	0.06	1.61 (1.16–2.22)	<0.001	1.43 (1.00–2.05)	0.05
Highest wealth percentile	1.43 (1.19–1.72)	<0.001	1.52 (1.18–1.94)	<0.001	1.63 (1.14–2.34)	0.01	1.74 (1.10–2.75)	0.02	1.60 (1.15–2.22)	<0.001	1.88 (1.20–2.93)	0.01	1.27 (0.93–1.73)	0.13	1.07 (0.70–1.63)	0.74
** *Health status* **																
Not sick	1		1		1		1		1		1		1		1	
Sick	1.47 (1.30–1.66)	<0.001	1.21 (1.06–1.38)	<0.001	1.09 (0.86–1.39)	0.48	0.95 (0.73–1.24)	0.71	1.85 (1.49–2.30)	<0.001	1.48 (1.17–1.87)	<0.001	1.47 (1.19–1.81)	<0.001	1.31 (1.05–1.64)	0.02
**(b) MDD**																
		**All ages**				**6–11 months**				**12–17 months**					**18–23 months**	
**MINIMUM DIETARY DIVERSITY**	**UNADJUSTED**		**ADJUSTED**		**UNADJUSTED**		**ADJUSTED**		**UNADJUSTED**		**ADJUSTED**		**UNADJUSTED**		**ADJUSTED**	
	**OR (95% CI)**	***p* value**	**OR (95% CI)**	***p* value**	**OR (95% CI)**	***p* value**	**OR (95% CI)**	***p* value**	**OR (95% CI)**	***p* value**	**OR (95% CI)**	***p* value**	**OR (95% CI)**	***p* value**	**OR (95% CI)**	***p* value**
** *Vaccination status* **																
Not vaccinated	1		1		1		1		1		1		1		1	
Partially vaccinated	4.28 (2.95–6.19)	<0.001	3.95 (2.69–5.80)	<0.001	2.86 (1.52–5.39)	<0.001	2.72 (1.34–5.49)	0.01	5.67 (2.69–11.93)	<0.001	4.14 (1.90–9.00)	<0.001	n/a	n/a	n/a	n/a
Fully vaccinated	3.82 (2.63–5.56)	<0.001	3.63 (2.45–5.41)	<0.001	3.13 (1.69–5.78)	<0.001	2.86 (1.43–5.71)	<0.001	6.15 (2.94–12.85)	<0.001	4.65 (2.15–10.00)	<0.001	n/a	n/a	n/a	n/a
** *Female empowerment* **																
Very low female empowerment	1		1		1		1		1		1		1		1	
Low female empowerment	1.02 (0.87–1.18)	0.83	0.98 (0.84–1.14)	0.80	0.79 (0.61–1.03)	0.08	0.80 (0.61–1.05)	0.10	0.96 (0.75–1.24)	0.75	0.94 (0.73–1.20)	0.69	1.24 (0.96–1.59)	0.11	1.38 (1.10–1.80)	0.02
Medium female empowerment	1.12 (0.94–1.34)	0.21	0.97 (0.82–1.15)	0.76	0.95 (0.70–1.28)	0.73	1.03 (0.75–1.41)	0.85	0.71 (0.52–0.96)	0.02	0.80 (0.58–1.10)	0.18	1.33 (0.99–1.77)	0.05	1.62 (1.20–2.20)	<0.001
High female empowerment	0.92 (0.66–1.29)	0.62	0.73 (0.53–1.02)	0.06	1.21 (0.71–2.04)	0.48	1.50 (0.87–2.58)	0.14	0.64 (0.37–1.11)	0.11	0.85 (0.48–1.50)	0.59	0.44 (0.22–0.86)	0.02	0.52 (0.30–1.04)	0.06
** *Wealth index* **																
First wealth percentile	1		1		1		1		1		1		1		1	
Second wealth percentile	1.43 (1.18–1.73)	<0.001	1.34 (1.18–1.53)	<0.001	1.17 (0.83–1.66)	0.36	1.16 (0.81–1.65)	0.42	1.84 (1.33–2.54)	<0.001	1.67 (1.20–2.30)	<0.001	1.29 (0.94–1.78)	0.12	1.21 (0.90–1.69)	0.27
Middle wealth percentile	1.54 (1.27–1.87)	<0.001	1.43 (1.17–1.75)	<0.001	1.41 (1.00–1.98)	0.05	1.34 (0.94–1.91)	0.10	1.47 (1.04–2.08)	0.03	1.32 (0.92–1.90)	0.14	1.77 (1.27–2.47)	<0.001	1.69 (1.20–2.39)	<0.001
Fourth wealth percentile	2.03 (1.66–2.47)	<0.001	1.80 (1.45–2.22)	<0.001	2.04 (1.44–2.89)	<0.001	1.86 (1.28–2.71)	<0.001	2.27 (1.61–3.19)	<0.001	1.93 (1.34–2.80)	<0.001	1.78 (1.27–2.50)	<0.001	1.66 (1.10–2.41)	0.01
Highest wealth percentile	2.51 (2.06–3.04)	<0.001	2.03 (1.58–2.60)	<0.001	2.17 (1.54–3.06)	<0.001	1.90 (1.23–2.92)	<0.001	3.34 (2.36–4.71)	<0.001	2.73 (1.74–4.30)	<0.001	2.20 (1.59–3.05)	<0.001	1.80 (1.20–2.77)	0.01
** *Health status* **																
Not sick	1		1		1		1		1		1		1		1	
Sick	1.34 (1.18–1.53)	<0.001	1.22 (1.06–1.41)	0.01	1.13 (0.89–1.44)	0.32	1.06 (0.82–1.37)	0.64	1.62 (1.28–2.06)	<0.001	1.46 (1.14–1.90)	<0.001	1.36 (1.09–1.69)	0.01	1.28 (1.00–1.61)	0.04

## Data Availability

All data generated or analysed during this study are included in this published article [and its Appendix A]. Primary data are available from the DHS archive.

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
