# Peer review of "Identifying Predictors for Minimum Dietary Diversity and Minimum Meal Frequency in Children Aged 6–23 Months in Uganda"

_nutrients, 2022, doi:10.3390/nu14245208_

Round 1
Reviewer 1 Report
This paper has good potential and would be much more interesting after some few issues are addressed. See comments in the attached

Author Response
Dear Reviewer,
We appreciated your time to review our manuscript. PLease find attached point-by point response.
Thank you.
Best,
Giulia Scarpa

Reviewer 2 Report
Abstract
1. On line 27-28, the authors write “Children who did not achieve 3 or more 28 complementary feeding indicators” but earlier they described that there were only 3 feeding indicators that were explored (MMF, MAD, MDD). Do you mean food groups?
Introduction
2. At the end of the 4th paragraph it says “Qualitative and mixed-methods research evaluating barriers of child feeding practices is also scarce [20].” However, this does not seem relevant since no qual data was collected or used in this paper.
Materials and Methods
3. Under indicators of complementary feeding when you mention the WHO guidelines for IYCF indicators, it would be good to add the date (2010) since there are now new guideline with modifications to the indicators (old MDD if 4 of 7 while new one Is 5 of 8 including breastmilk) so it is important to highlight.
4. For table 3, it would be helpful to add another column to explain the justification behind each of the hypotheses.
5. There is some repetition in the methods section for example related to presentation of the covariates.
Results
6. Figure 1 is missing some labels. The middle set of bars is not labeled and it is unclear where MAD is. This figure requires some formatting.
7. Can you please clarify the timeframe of assessment for the exposure variables -- illness two weeks prior and vaccination status? At what ages was morbidity and vaccination status assessed?
8. Why did you choose to calculate and report ORs instead of RRs?
Discussion
9. It is interesting that you found that illness increased the odds of achieving the IYCF indicators despite likely changes in appetite. I wonder if the finding of no association with the 6-11 month outcomes is because of the prevalence of breastmilk feeding?
10. I am not sure that the argument that children who are vaccinated are less likely to be sick is supporting the association between vaccination and IYCF outcomes given that you found the opposite association with illness. Suggest removing or reworking this sentence on page 17, lines 449-450.
Conclusion
11. I suggest tailoring the statements in the introduction to be related more to how the exposures/risk factors can be changes in order to better achieve the IYCF indicators since it is currently too broad.
Author Response
Dear Reviewer,
We appreciated your time to comment on our manuscript. Please find attached a point-by-point response.
Thank you!
Best,
Giulia Scarpa
